# The Role of α-Synuclein in the Regulation of Serotonin System: Physiological and Pathological Features

**DOI:** 10.3390/biomedicines11020541

**Published:** 2023-02-13

**Authors:** Lluis Miquel-Rio, Unai Sarriés-Serrano, Rubén Pavia-Collado, J Javier Meana, Analia Bortolozzi

**Affiliations:** 1Institute of Biomedical Research of Barcelona (IIBB), Spanish National Research Council (CSIC), 08036 Barcelona, Spain; 2Institut d’Investigacions Biomèdiques August Pi i Sunyer (IDIBAPS), 08036 Barcelona, Spain; 3Biomedical Research Networking Center for Mental Health (CIBERSAM), Institute of Health Carlos III (ISCIII), 28029 Madrid, Spain; 4Faculty of Medicine and Health Sciences, University of Barcelona (UB), 08036 Barcelona, Spain; 5Department of Pharmacology, University of the Basque Country UPV/EHU, 48940 Leioa, Spain; 6MiCure Therapeutics Ltd., Tel Aviv 6423902, Israel; 7Department of Pharmacology, CIBERSAM, Biocruces Bizkaia Health Research Institute, University of the Basque Country UPV/EHU, 48940 Leioa, Spain; 8Biocruces Bizkaia Health Research Institute, 48940 Leioa, Spain

**Keywords:** depression, Parkinson’s disease, α-synuclein, serotonin, raphe nuclei

## Abstract

In patients affected by Parkinson’s disease (PD), up to 50% of them experience cognitive changes, and psychiatric disturbances, such as anxiety and depression, often precede the onset of motor symptoms and have a negative impact on their quality of life. Pathologically, PD is characterized by the loss of dopamine (DA) neurons in the substantia nigra pars compacta (SNc) and the presence of intracellular inclusions, called Lewy bodies and Lewy neurites, composed mostly of α-synuclein (α-Syn). Much of PD research has focused on the role of α-Syn aggregates in the degeneration of SNc DA neurons due to the impact of striatal DA deficits on classical motor phenotypes. However, abundant Lewy pathology is also found in other brain regions including the midbrain raphe nuclei, which may contribute to non-motor symptoms. Indeed, dysfunction of the serotonergic (5-HT) system, which regulates mood and emotional pathways, occurs during the premotor phase of PD. However, little is known about the functional consequences of α-Syn inclusions in this neuronal population other than DA neurons. Here, we provide an overview of the current knowledge of α-Syn and its role in regulating the 5-HT function in health and disease. Understanding the relative contributions to α-Syn-linked alterations in the 5-HT system may provide a basis for identifying PD patients at risk for developing depression and could lead to a more targeted therapeutic approach.

## 1. Introduction

Parkinson’s disease (PD) is clinically characterized based on classic motor features including the presence of hypokinesia, rigidity, resting tremor, and impaired postural control [1,2,3]. A wide variety of incapacitating non-motor symptoms are also present over the course of the illness. These non-motor signs include autonomic and neuropsychiatric features such as fatigue, apathy, anxiety and depression, as well as cognitive deficits. Neuropsychiatric symptoms are inherent to the disease and are neither a result nor a side effect of long-term dopaminergic treatment [4,5]. These comorbidities are frequent and can be found in all stages of PD, from the premotor and the early untreated phases of the disease to the advanced stages of PD [6,7,8,9,10,11,12,13]. Among them, depression is one of the most prevalent neuropsychiatric symptoms, ranging from 35 to 50% of patients with PD [14,15,16]. Depressive disorder represents a huge burden on the quality of life in many PD patients, but is frequently undiagnosed and left untreated [17,18,19,20]. Therefore, understanding the neurobiology of depression in PD is critical to achieving the optimal care needed by patients with PD.

While the etiology of PD still remains unclear, one major neuropathological hallmark of PD is the degeneration and subsequent loss of DA neurons in the substantia nigra pars compacta (SNc) leading to prototypic motor deficits [21,22,23]. The SNc involves a neuronal population projecting to the caudate and putamen and is critical for the regulation of basal ganglia circuitry [24,25]. Lewy pathology (LP), which can also be observed across the central, peripheral, and enteric nervous systems (CNS, PNS, and ENS), is another major pathological finding present in about 70% of “clinically typical PD cases” [26,27]. This includes both Lewy bodies (LB) and Lewy neurites (LN), which are composed of a variety of different molecules, proteins, and organelles, including ubiquitin, tubulin, neurofilaments, lipids, and mitochondria. Among them, aggregates of α-synuclein (α-Syn) protein represent one of the main LP components [28,29,30,31,32]. To explain the widespread localization of LP and the onset of the various non-motor symptoms of PD, a critical point to consider is the dysfunction of other neuronal populations and neurotransmitter systems in regions of the CNS and PNS, other than the SNc DA neurons. Indeed, several studies reported LB-associated deficits—most likely occurring even prior to DA neurons—in cholinergic neurons in the pedunculopontine nucleus, nucleus basalis of Meynert and of the dorsal motor nucleus of the vagus, as well as in norepinephrine—NE neurons of the locus coeruleus (LC), and serotonin—5-HT (5-hydroxytryptamine) neurons of the raphe nuclei (RN) [33,34,35]. Furthermore, altered GABAergic and glutamatergic signaling was also reported in the amygdala and several cortical brain regions that may play important roles in the complex cognitive features of PD [34,36,37]. 

Lately, attention has been focused on the impaired integrity of the 5-HT system in PD, in addition to its well-known role in the pathogenesis of anxiety and depressive disorders. Notably, a growing amount of research supports a specific causal role of 5-HT system dysfunction in the progression of several PD symptoms, such as tremor and dyskinesia, but also anxiety and depression at early stages of the disease [5,38,39,40,41,42,43]. This review will discuss recent findings of the role of α-Syn in regulating the 5-HT system. A better understanding of the relative contributions of α-Syn-related abnormalities of the 5-HT system could lead to the identification of PD patients who are at risk of developing depression, as well as to better animal models of the disease and a more tailored therapeutic approach.

## 2. Connectivity of the Brain Serotonin System

A complete review of the brain 5-HT system is beyond the scope of the present article. The reader is referred to several reviews in the literature [44,45,46]. Here, we would like to highlight some features of the connectivity of the 5-HT system directly linked to its role in the neurobiology of depression. 

The brain 5-HT system exerts its widespread effects from a group of relatively small brainstem nuclei known as the RN. Raphe 5-HT-producing neurons send ascending projections to the entire brain as well as descending projections to the spinal cord [47] (Figure 1A). These projections form classical synaptic connections, as well as varicosities with no associated postsynaptic structure [48,49]. Upon release, 5-HT acts primarily on G-protein coupled receptors (5-HT_1_, 5-HT_2_, 5-HT_4_, 5-HT_5_, 5-HT_6_, 5-HT_7_, and a single ionotropic receptor 5-HT_3_) encoded by more than a dozen distinct genes and many more isoforms, which are differentially expressed in the brain [50,51]. Indeed, all brain regions express multiple 5-HT receptors in a receptor subtype-specific pattern [52]. In addition, individual neurons may express several 5-HT receptor subtypes. For instance, pyramidal neurons in layer V of the ventromedial prefrontal cortex (vmPFC) express 5-HT_1A_ and 5-HT_2A_ receptors, which exert opposite effects on neuronal firing activity [53,54]. Hence, the plethora of effects of the brain’s 5-HT system is partly explained by the fact that 5-HT neurons are optimally positioned to affect the activity of a wide range of brain networks.

Among the different raphe nuclei, the dorsal raphe nucleus (DR) is the largest serotonergic nucleus, containing approximately one third of all 5-HT neurons in the brain [47]. As such, the human DR comprises about 250,000 neurons, out of a total of 10^11^ neurons in the whole brain—approximately 20,000 5-HT-producing neurons in the rat—and its axons branch widely, innervating almost all brain areas. This can be illustrated in the rat cortex, where >10^6^ serotonergic nerve endings/mm^3^ were noted. In addition, each cortical neuron may receive around 200 varicosities [55]. Unlike cortical and subcortical glutamatergic projection neurons that exhibit precise short- or long-distance connectivity with other neuronal groups [56], DR 5-HT cells send highly divergent ascending projections connecting brain areas with different functions [57]. Indeed, correlations have been reported between changes in DR 5-HT neuron activity and different cognitive processes, such as working memory [58], cognitive flexibility [59], response inhibition [60], and exploration–exploitation balance [61]. Furthermore, the raphe 5-HT system is also involved in the modulation of mood, emotion, perception, stress, reward, aggression, and social interactions, among others [62,63,64,65,66]. It is difficult to find a human behavior that is not regulated by a 5-HT response.

Notably, deficits in the 5-HT signaling are implicated in the neuropathology of anxiety and depression. Imbalances in the production and transmission of several neurotransmitters, including 5-HT, are commonly observed in the CNS of patients suffering from depressive disorder [67]. In fact, a widely accepted etiological theory is the “monoamine hypothesis of depression”, which postulates that depression disorder is associated with a decreased monoamine function (NE, DA, and 5-HT) in key brain areas, such as the vmPFC, hippocampus (HPC), amygdala (AMG), nucleus accumbens (NAc), ventral tegmental area (VTA), and hypothalamus [68,69,70]. Notably, neuroimaging studies associate vmPFC with a broad spectrum ranging from emotion to cognitive functions, and alterations in vmPFC activity have been correlated with the biology of depression as with favorable outcomes of novel antidepressant strategies [71,72,73]. Likewise, structural and functional neuroimaging studies show pronounced alterations in vmPFC circuits in patients with depression and PD [74,75,76]. The vmPFC, which is composed of 75–80% glutamatergic pyramidal projection neurons and 20–25% GABAergic local circuit interneurons, is strongly innervated by DR 5-HT neurons [53,54]. The 5-HT fibers exert an important modulatory role of excitatory and inhibitory currents in vmPFC neurons [77,78], mainly through activation of 5-HT_2A_ and 5-HT_1A_ receptors, respectively. In turn, the monoamine groups, including the 5-HT neurons of the DR, are innervated by descending axons from layer V pyramidal neurons in the vmPFC [79] that control the monoamine neuron activity [80,81], thus establishing a reciprocal connectivity and mutual control (Figure 1B). Although it is beyond the scope of this review, ultimately, one can be optimistic that the functional integrity of the vmPFC–raphe nuclei circuit will play an important role in the pathophysiology of depression in early PD. New approaches are advancing in many directions to identify early PD, and the 5-HT system and its connections are an important part of these recent efforts, as will be described in the following sessions. Advanced neuroimaging techniques, next generation RNA sequencing, the recent addition of the proximity ligation assay (PLA) that specifically recognizes α-Syn aggregates and new animal models, among others, will provide support for the classification of PD based on different pathological phenotypes, leading to a more appropriate therapeutic strategy. 

## 3. α-Synuclein and Serotonin Neurotransmission

α-Syn is a small, natively unfolded protein belonging to the synuclein family that also encompasses β-synuclein (β-Syn) and γ-synuclein (γ-Syn). These are evolutionarily conserved proteins that have currently only been described in vertebrates, supporting the notion that they regulate some essential physiological functions [82,83,84,85]. Between them, α-Syn is the most studied protein of this family, due to its crucial role in the pathogenesis of PD and other synucleinopathies [86]. This protein is characterized by a remarkable conformational plasticity, adopting different conformations depending on the environment, i.e., neighboring proteins, lipid membranes, redox state, and local pH [87,88,89,90]. In fact, α-Syn adopts a monomeric, random coil conformation in an aqueous solution, while its interaction with lipid membranes drives the transition of the molecule part into α-helical structure. The central unstructured region of α-Syn is involved in fibril formation by converting to well-defined, β-sheet rich secondary structures. These structural and biophysical properties probably hold the key to their normal and abnormal function [91,92]. α-Syn is abundantly expressed in all neuronal types, where it localizes in presynaptic terminals [93,94,95] and modulates synaptic functions [96,97,98]. However, α-Syn is among the last presynaptic proteins to become enriched at the synapse [94] and unlike γ-Syn, it does not seem to be involved in synaptic development [99,100]. Recent studies have revealed that α-Syn is also present in different organelles, including nuclei, mitochondria, Golgi, and endoplasmic reticulum (ER) [93,101,102,103], although in lower concentrations than those found in synaptic locations, and its function is even less well understood [84]. This feature makes α-Syn a hub within synaptic protein interaction networks [84]. Supporting this, α-Syn was first identified at the presynaptic level as interacting with synaptic vesicle (SV)-associated proteins [93]. Indeed, it cooperates with a large number of SV surface proteins including the synapsin phosphoprotein family, complexins, and mammalian Munc 13-1, described to be affected in brain samples from PD patients and in various human α-Syn transgenic mouse lines [98,99,104,105,106]. Furthermore, several studies also indicated that α-Syn interacts with the SV glycoprotein 2 (SV2) family to positively modulate vesicular functions in a variety of ways, possibly by aiding in vesicular trafficking and exocytosis, as well as stabilizing stored transmitters [107,108]. In this regard, both postmortem PD brain tissue and animals overexpressing mutant α-Syn showed increases in the SV2C protein, which is abundantly expressed in the basal ganglia and selectively localizes to DA neurons [109]. Similarly, elevated levels of SV2A protein co-localizing with α-Syn were found in axonal swellings across the caudate-putamen (CPu) and cingulate cortex in a mouse model overexpressing human wild-type α-Syn in 5-HT neurons [110]. Other proteins such as Rabs, which in addition to modulating axonal traffic are also very important for the regulation of each step leading to SV release, docking and fusion at synaptic sites, interact with α-Syn [111,112]. Actually, several findings support that Rabs play a crucial role as direct mediators in the induction of synaptic alterations concerning α-Syn leading to PD pathology [112]. Taken together, the loss of α-Syn function, coupled with changes in its levels at synaptic terminals, can cause multifaceted dysregulation of many other synaptic proteins involved in neurotransmission mechanisms. 

In addition to being involved in synaptic vesicular trafficking, α-Syn is also directly engaged in the regulation of monoamine (DA, NE, and 5-HT) neurotransmission homeostasis—β-Syn and γ-Syn are also involved in this regulation, although their role is less known [85,113,114,115,116,117,118]. Monoamine transporters (MAT) are transmembrane proteins solely responsible for the synaptic reuptake of DA, NE and 5-HT, and partly maintain the homeostasis of monoaminergic neurotransmission. MAT are important pharmacological targets in the therapy of various neuropsychiatric diseases, such as anxiety, depression, and suicidal behavior, among others, due to their crucial role within the brain in the replacement of monoamine neurotransmitters [119,120]. Direct interactions between α-Syn and MAT proteins have been described, indicating an important role for the synucleins in regulating MAT function, trafficking and distribution at the synapse. Even though most of the evidence is focused on DA neurotransmission and its transporter (DAT), in this review we will emphasize the role of α-Syn in the homeostasis of 5-HT neurotransmission.

Previous studies showed that the cell-surface expression and function of the 5-HT transporter (SERT) in co-transfected cells are negatively modulated by α-Syn in a non-Abeta-amyloid component (NAC) domain-dependent manner [115]. In addition, pioneering reports also showed direct interactions of α-Syn-SERT and γ-Syn-SERT proteins in cultured cells and in rat brain tissue, assessed by immunoprecipitation [115,121]. α-Syn-induced modulation of SERT trafficking is microtubule-dependent, as the microtubule-destabilizing agent nocodazole disrupts the effects of α-Syn on SERT function, reversing the inhibition of uptake in co-transfected cells [116]. More recently, in vivo studies indicated that down-regulation of α-Syn expression in raphe 5-HT neurons induced by an antisense oligonucleotide (ASO) leaves an increased synaptic 5-HT concentration, which was dependent on the reduction of SERT activity, as assessed by the selective SERT inhibitor citalopram [118]. The overexpression of α-Syn in raphe nuclei produced the opposite effects, with mice exhibiting a drop in extracellular 5-HT levels that was dependent on SERT function [110].

Moreover, α-Syn is also involved in the vesicular storage of monoamine neurotransmitters by the vesicular monoamine transporter 2 (VMAT_2_). VMAT_2_ mobilizes monoamines from the neuronal cytoplasm into vesicles, where they are repackaged for release at synapses [122,123]. VMAT_2_ co-localizes with α-Syn protein in the Lewy bodies from PD brains [124], and overexpression of α-Syn negatively impairs VMAT_2_ expression/function, leading to increased levels of cytosolic monoamine in presynaptic terminals, which in turn induce neurotoxicity [113]. These findings suggest that α-Syn may maintain high VMAT_2_ activity to protect monoamine neurons form cell death [125]. In support of this view, in vivo studies showed that down-regulation of α-Syn expression in DA and 5-HT neurons increases the releasable pool of DA and 5-HT sensitive to tetrabenazine, a selective inhibitor of VMAT_2_ [118]. Overall, the presynaptic location of α-Syn has suggested a physiological role in neurotransmitter release and it apparently associates with the SV clustering and storage [98,99]. Furthermore, α-Syn is abundantly expressed in DA, NA, and 5-HT neurons [96,118], defining a precise role of α-Syn in monoamine synaptic plasticity by interacting with specific proteins that maintain monoamine homeostasis. 

## 4. Dysfunction of the 5-HT System in PD Patients 

The investigation of premotor pathology presents one of the most difficult problems in PD research. Although Braak and colleagues [26,126] proposed a significant premotor phase that may last as long as the symptomatic period, the identification of this phase in clinical practice is elusive. In fact, the profile of PD patients is also associated with diverse symptoms and clinical phenotypes [127]. Cumulative evidence indicates the existence of ongoing pre-SNc DA neurodegeneration during the premotor phase leading to non-motor symptoms, mainly constipation, anxiety and depression, smell loss, and rapid-eye-movement (REM) sleep behavior disorder [128]. A dysfunctional 5-HT system is generally regarded as a risk factor for depression. Consistent with this view, several reports suggest a positive correlation between decreased 5-HT neurotransmission and the severity of depression and anxiety symptoms in PD, most likely caused by pathological changes of the 5-HT neurons in the midbrain raphe nuclei [39,41,43,129,130]. 

By evaluating SERT availability with positron emission tomography (PET) and single photon emission computed tomography (SPECT) scans using various radioactive ligands, one can assess the integrity of the 5-HT system. The non-specific ligands [^123^I]β-CIT and [^123^I]FP-CIT have mostly been employed in in vivo SPECT imaging. Although these ligands have similar affinities for DAT and SERT, their thalamic and midbrain binding are considered to be SERT-specific [131]. Hence, SPECT studies using [^123^I]β-CIT and [^123^I]FP-CIT found decreased binding in the thalamus and midbrain of PD patients [132,133,134,135,136]. The PET ligands [^11^C]-DASB and [^11^C](+)McN5652 are highly specific for SERT. Using these ligands, several reports indicated reduced binding in different brain regions including the frontal cortex, striatum, and raphe nuclei [137,138,139]. Interestingly, an early study using [^11^C]-DASB to map SERT changes in various PD stages based on disease duration showed reduced binding in the striatum, thalamus and anterior cingulate cortex of early-stage PD patients [140]. In the same study, decreases in SERT binding were observed in the prefrontal cortex of established PD and in the rostral and caudal raphe nuclei in advanced stages [140]. Moreover, recent SPECT and PET studies also showed 5-HT pathology in the premotor phase in mutant A53T α-Syn gene *(SNCA)* carriers, before striatal DA loss, highlighting the early role of 5-HT pathology in the progression of PD [130]. 

The aforementioned studies examined PD patients without depressive symptoms. However, numerous investigations have assessed the connection between depression and SERT binding in PD. In a small cohort of depressed PD patients, early studies with [^11^C]-DASB PET demonstrated that depression correlated with increased SERT binding in the dorso-lateral and prefrontal cortex [39]. Other studies also reported that the cingulate cortex and caudal raphe nuclei of depressed PD patients showed higher levels of SERT than non-depressed PD patients [41,141]. These findings are in agreement with the low levels of 5-HT and its metabolite 5-hydroxyindoleacetic acid (5-HIAA) found in the cerebrospinal fluid (CSF) of patients with PD and depression [142]. Interestingly, CSF levels of homovanillic acid (HVA), a DA metabolite, were not associated with the presence of depression in PD [142]. Likewise, in response to the functional deficit of 5-HT availability, post-synaptic 5-HT_1A_ and 5-HT_2A_ receptors were upregulated in cortical brain regions [143]. Furthermore, the density of 5-HT_2C_ and 5-HT_2A_ receptors in SN pars reticulate and striatum, respectively, appears to be increased in patients with PD [144,145,146]. These alterations may represent a compensatory response to a reduction of functional 5-HT levels in these nuclei. 

In contrast, more recent PET studies revealed that the severe apathy in PD patients correlates with a reduction of [^11^C]-DASB binding in the anterior caudate nucleus and orbitofrontal cortex, while the depression degree was exclusively related to a reduction in [^11^C]-DASB binding within the bilateral subgenual anterior cingulate cortex (ACC) [5]. In another SPECT study, [^123^I]FP-CIT binding was decreased in the midbrain of a cohort of PD patients with depression [147]. In light of this, it appears that, although imaging studies indicate that SERT binding in the midbrain and forebrain differs between non-depressed and depressed patients with PD, the extent to which these changes are crucial for the onset of depression is still unknown. Data also suggest that the presence of 5-HT pathology occurs at the beginning of the disease, preceding the development of the DA pathology and motor symptoms. Therefore, molecular imaging of SERT could be used to visualize the premotor pathology of PD in vivo as an adjunctive tool for screening and monitoring progression for individuals at risk of PD, thereby complementing DA imaging. 

Importantly, neuropathological studies have demonstrated the presence of LBs (α-Syn positive staining) in raphe 5-HT neurons in the early stages of the disease [148,149,150,151]. Previous studies on the propagation of α-Syn proposed that PD begins in the medulla oblongata with LB pathology in the dorsal motor nuclei of the glossopharyngeal and vagal nerves and the adjacent intermediate reticular zone [126,152]. As PD progresses, it is proposed that the LB pathology spreads up the brainstem in an upward direction, affecting the raphe nuclei before reaching the SNc. In late stages, LBs are also found in limbic and cortical brain areas. The caudal groups of the raphe nuclei (e.g., raphe major, raphe obscure, and raphe pallidus) have been widely shown to contain LB-related lesions in the early stages of PD or even before the onset of motor symptoms [126,152,153]. The 5-HT neurons found in the caudal raphe nuclei play a role in a number of autonomic processes, including pain and decreased gastrointestinal motility, which are recognized non-motor symptoms in PD. In addition, the rostral raphe nuclei, containing the DR and median raphe nucleus (MR), also appear to be affected in PD. LB pathologies have been found in both the DR and MR of post-mortem PD brains and appear to be localized in 5-HT-containing neurons [148,149,154]. Some early studies found a significant neuronal loss within the DR from postmortem brain samples of PD patients, and this was even more pronounced in depressed PD patients [155]. However, other studies did not observe neuronal loss in DR, but did in MR [148,149,156]. Surprisingly, we found that only one of these studies used an unbiased design-based stereology method for counting cells [156]. In addition, some studies reported the absence of significant cell loss in the DR of post-mortem PD brains, but found evidence of the dysfunction of DR neurons based on reduced nucleolar volume and loss of cytoplasmic RNA [157]. Recently, it was also reported that long-range 5-HT projections from raphe are vulnerable in PD in response to hydrogen peroxide-induced cellular stress [158]. Taken together, the above findings point to neuropathological alterations in the 5-HT system in PD, comprising of the presence of LB pathology accompanied in some cases by neuronal loss in the raphe nuclei, as well as morphological changes of 5-HT fibers, which would lead to modified 5-HT neurotransmission.

## 5. Dysfunction of the 5-HT System in Animal Models with Overexpression of α-Syn

Abundant evidence suggests that the development of PD may comprise three main phases. The onset of α-Syn buildup in the CNS or PNS/ENS, in the absence of observable clinical symptoms, is referred to as the “preclinical PD” phase. The second phase, often known as the “pre-motor” or “prodromal,” can last for more than 10 years before the disease is clinically diagnosed. It is usually accompanied by the appearance of non-motor symptoms caused in part by pre-SNc abnormalities. During this phase, PD patients may display increased anxiety as early as 16 years prior to disease diagnosis; and depression becomes significantly prevalent among PD patients in the last 3–4 years preceding diagnosis. The third phase is the “motor phase of PD”, which is the one that is clinically visible and easiest to diagnose [159,160]. Understanding the pathophysiological mechanisms underlying non-motor symptoms in PD is important, but requires relevant preclinical animal models. In this sense, one of the main shortcomings of current PD-like animal models is that they focus on DA pathways, which probably do not reflect the complexity underlying the occurrence of these symptoms in patients [161,162]. In fact, there is still a paucity of studies addressing the role of brain circuits other than nigrostriatal DA systems in the early stages of the disease. In this review, we provide an overview of the current state of the field by presenting different preclinical models used in research on measures of anxiety and depressive phenotypes in rodent models of PD with emphasis on 5-HT systems.

In addition to the toxin-induced and genetic animal models of PD [162,163,164,165,166,167], in recent decades, an alternative approach to modulate the disease based on the forced expression of wild-type or mutant human α-Syn using (1) transgenic techniques, (2) viral vector mediated transfer of α-Syn, or (3) injection of pathogenic pre-formed α-Syn fibrils (PFFs) has been presented. Thus, an intracellular accumulation of α-Syn in raphe 5-HT neurons and in hippocampal 5-HT fibers, without loss of 5-HT neurons in 12-week-old transgenic mice overexpressing mutant A53T α-Syn, was reported [168]. In parallel, mice showed a reduced 5-HT release and compromised increase in doublecortin+ neuroblasts in the dentate gyrus (DG), indicating a differential neurogenic response [168]. Another study also reported that mutant A53T α-Syn–expressing mice (52-week-old) showed strong α-Syn expression in the prefrontal cortex (PFC) along with reduced 5-HT innervation in layers V/VI of the PFC and enlarged axonal varicosities [169], leading to altered 5-HT signaling. 

Moreover, Khol et al. [170] generated an α-Syn transgenic rat model that displayed important features of PD such as a widespread and progressive α-Syn aggregation pathology, DA loss and age-dependent motor decline. Notably, prior to the occurrence of the motor phenotype, rats showed profoundly impaired dendritogenesis of neuroblasts in the hippocampal DG, resulting in the severely reduced survival of adult newborn neurons. Reduced 5-HT_1B_ receptor levels, lower 5-HT neurotransmitter concentration, and loss of 5-HT nerve terminals innervating the DG/CA3 subfield were indicative of decreased neurogenesis, but the number of 5-HT neurons in the raphe nuclei remained stable. The authors highlight that this transgenic rat model elicited an early anxiety-like phenotype, suggesting that α-Syn accumulation severely impairs hippocampal neurogenesis and 5-HT neurotransmission prior to motor function [170].

Recently, the adeno-associated virus (AAV)-α-Syn and PFFs models have been specifically adapted for study of α-synucleinopathies using stereotaxic delivery into different brain areas, making them useful tools [171]. Therefore, a model of AAV-induced α-synucleinopathy selectively in 5-HT neurons of rats resulted in progressive degeneration of the 5-HT axon terminals in hippocampus, without the loss of raphe 5-HT neurons [172]. Furthermore, overexpression of α-Syn in raphe nuclei and basal forebrain cholinergic neurons of rats resulted in a more pronounced axonal pathology and significantly impaired anxiety response as assessed in the elevated plus maze [172]. Likewise, we demonstrated that AAV-induced overexpression of human α-Syn in mouse 5-HT neurons causes a gradual accumulation and aggregation of α-Syn in the 5-HT system. In parallel, we found alterations in axonal transport, brain-derived neurotrophic factor (BDNF) production, and 5-HT neurotransmission in 5-HT projection brain areas of PD-like mice, leading to a depressive-like phenotype (Figure 2) [110]. 

Other studies using rodent models of PFF also reported motor deficits and emotional and cognitive abnormalities, although the latter findings remained relatively unchanged in PFF models up to 6 months after injection [173,174,175]. Whether this is due to the injection site (primarily such as the striatum, SNc, or olfactory bulb) and/or the duration of the pathological spread remains to be determined. For instance, a recent study found that PFF injections in mice caused deficits in social dominance behavior and fear conditioning, two activities linked to prefrontal cortex and amygdala function, suggesting that these brain regions may be crucial in the complex emotional and cognitive traits of PD [37].

In addition, some studies using cell cultures overexpressing α-Syn showed that 5-hydroxyindoleacetaldehyde (5-HIAL), a 5-HT metabolite product generated by monoamine oxidase (MAO-A), increases α-Syn oligomerization, which may explain the dysfunction of 5-HT neurons in PD [176]. Recent studies also showed the importance of maintaining the integrity of 5-HT systems, as 5-HT itself can affect the growth of amyloid-forming protein fibrils. Indeed, 5-HT or selective serotonin reuptake inhibitors (e.g., escitalopram) activate signaling that alters the processing of α-Syn fibrils as well as amyloid precursor proteins into β-amyloid (Aβ) to prevent protein aggregation by direct binding, and could be beneficial to PD and other neurodegenerative disorders [177,178,179]. 

## 6. Conclusions

The frequent occurrence of depression in PD is a prevalent and complex issue. Although often overlooked or underestimated, depression can seriously influence the course of PD and the quality of life of patients. In addition to dopaminergic depletion, several findings highlight the importance of serotonergic degeneration in PD. Thus, changes in 5-HT biochemical markers, LB pathology (α-Syn-positive staining) in raphe nuclei, and structural and functional alterations in the serotonergic system have been described, and it has been shown that these alterations in the serotonergic connectome are mainly associated with the expression of neuropsychiatric symptoms at disease onset. In support of this, the few available animal models demonstrating α-Syn-induced deficits in the serotonergic system recapitulate the mechanisms and early premotor stages of the disease. Altogether, measuring serotonergic integrity might be a useful in vivo tool to use in routines to guide the choice of the pharmacological arsenal in order to alleviate PD-related neuropsychiatric symptoms. Thus, such a measurement could serve as a sensitive marker of PD burden. 

## Figures and Tables

**Figure 1 biomedicines-11-00541-f001:**
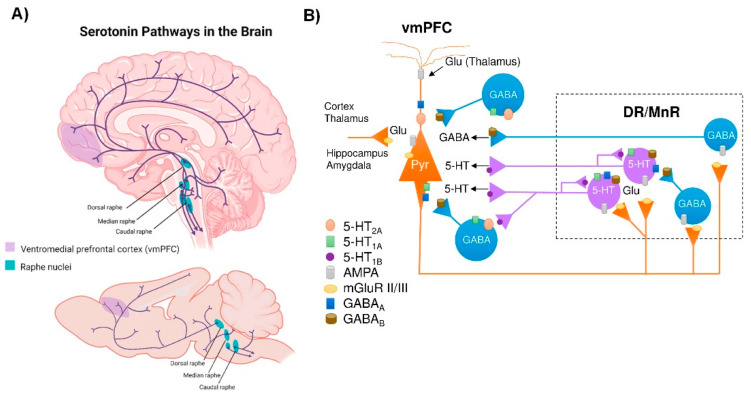
Central serotonergic pathways. (**A**) Schematic representation of the raphe nuclei in humans (**top**) and mice (**bottom**), which give rise to ascending projections to large regions of the brain, as well as descending projections predominantly innervate the cerebellum and its input structures and to the spinal cord. (**B**) Diagram showing how the ventromedial prefrontal cortex (vmPFC) and the dorsal and median raphe nuclei (DR and MrR, respectively) are anatomically and functionally connected in both directions. Pyramidal glutamatergic neurons from vmPFC send axons to raphe nuclei, where they form excitatory synapses (AMPA receptors) with 5-HT and GABAergic neurons. Stimulation of glutamatergic neurons in vmPFC primarily triggers inhibitory responses in 5-HT neurons mediated by (i) the activation of local GABAergic circuits that control the activity of 5-HT neurons in the raphe nuclei and (ii) 5-HT_1A_ autoreceptor-dependent self-inhibitory responses following excitatory activation of 5-HT neurons. In addition, DR/MnR 5-HT neurons control the activity of glutamatergic neurons in the vmPFC through inhibitory 5-HT_1A_ receptors and excitatory 5-HT_2A_ receptors expressed in glutamatergic and GABAergic neurons. Similarly, the activity of the vmPFC-DR/MnR pathway may be affected by the activation of 5-HT_4_ receptors on glutamatergic neurons and 5-HT_3_ receptors on GABAergic interneurons in the outer layer of the vmPFC (not shown in the diagram). Adapted from [53,54,78].

**Figure 2 biomedicines-11-00541-f002:**
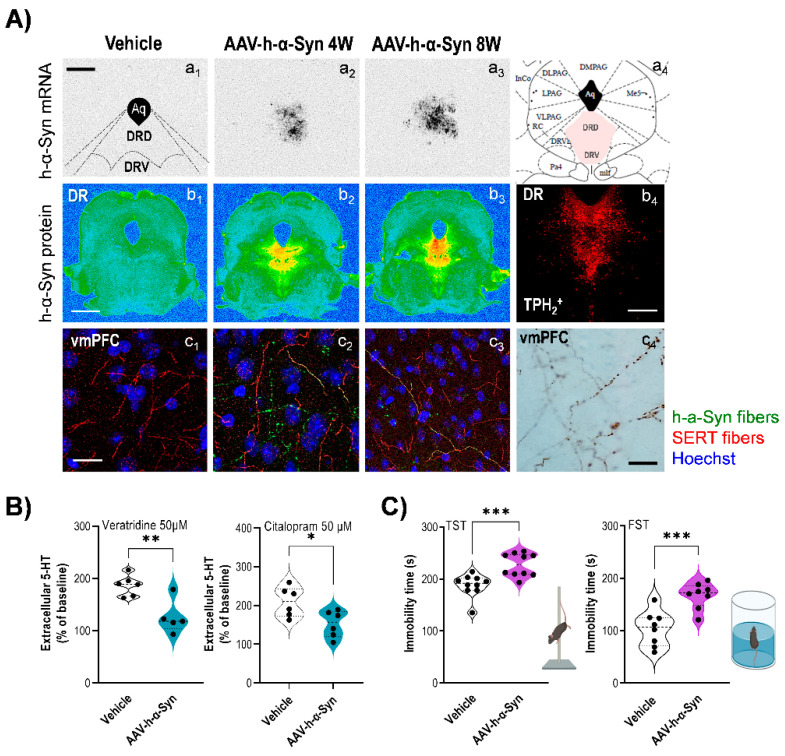
Human α-synuclein (h-α-Syn) overexpression in mouse serotonin neurons triggers a depressive-like phenotype. H-α-Syn expression was driven to raphe nuclei by a 1 μL AAV construct harboring a chicken-actin promoter (AAV-h-α-Syn) or vehicle, and mice were killed 1, 4, and 8 weeks (W) after injection. (**A**) Representative midbrain sections showing h-α-Syn mRNA levels in raphe nuclei examined by an in situ hybridization technique (**a_1_**–**a_3_**). Schematic coronal representation of mouse midbrain at −4.72 mm (AP coordinate) from bregma (**a_4_**). Scale bar: 500 μm. Abbreviations: Aq (aqueduct), DRD (dorsal raphe nucleus, dorsal), and DRV (dorsal raphe nucleus, ventral). Representative coronal midbrain sections showing progressive increases of h-α-Syn protein levels in the raphe nuclei assessed by immunohistochemistry procedures (**b_1_**–**b_3_**). Signal represents the optical density (OD) of autoradiograms. Scale bar: 1 mm. Raphe serotonin (5-HT) neurons were identify using tryptophan hydroxylase (TPH_2_) marker (**b_4_**). Scale bar: 25 μm. Representative confocal microscopy images showing serotonin transporter (SERT) and h-α-Syn axonal co-localization in different ventromedial prefrontal cortex (vmPFC) of mice injected with AAV5 examined 4 W and 8 W later (**c_1_**–**c_3_**). The majority of the h-α-Syn-positive fibers also showed SERT staining, proving that they originate from raphe nuclei. Scale bar: 25 μm. Immunohistochemistry procedure on representative coronal brain sections reveals h-α-Syn-positive axonal swellings in the vmPFC (**c_4_**). Scale bar: 25 μm. (**B**) Local infusion of veratridine (depolarizing agent, 50 μM) or citalopram (selective serotonin transporter inhibitor 50 μM) into vmPFC induced a greater effect on 5-HT release in vehicle-injected than in AAV-h-α-Syn-injected mice at 4 W post-administration. (**C**) AAV-injected mice evoked a depressive-like state in the tail suspension (TST) and forced swimming (FST) tests characterized by a longer immobility time compared to vehicle-injected mice. Values are presented as mean ± SEM. * *p* < 0.05, ** *p* < 0.01, and *** *p* < 0.001, compared to vehicle-injected mice. Adapted from [110].

## Data Availability

Not applicable.

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
