# Peer review of "The Role of α-Synuclein in the Regulation of Serotonin System: Physiological and Pathological Features"

_biomedicines, 2023, doi:10.3390/biomedicines11020541_

Round 1
Reviewer 1 Report
The authors want to provide an overview of current knowledge of α-Syn and their role in regulating the 5-HT function in health and disease and provide a basis for identifying PD patients at risk for developing depression and could lead to a more targeted therapeutic approach.
1. α-Syn is a small, natively unfolded protein belonging to the synuclein family that also encompasses β-synuclein (β-Syn) and γ-synuclein (γ-Syn).These are evolutionarily conserved proteins which have currently only been described in vertebrates supporting the notion that they regulate some essential physiological functions. In order to understand the physiological function, the author should describe the synthetic and metabolic pathways of α-Syn.
2. The authors should describe the difference between physiological α-Syn and pathological α-Syn(mutant α-Syn) in detail.
Author Response
- α-Syn is a small, natively unfolded protein belonging to the synuclein family that also encompasses β-synuclein (β-Syn) and γ-synuclein (γ-Syn).These are evolutionarily conserved proteins which have currently only been described in vertebrates supporting the notion that they regulate some essential physiological functions. In order to understand the physiological function, the author should describe the synthetic and metabolic pathways of α-Syn.
We appreciate this reviewer's comments on the study and his/her interest in understanding in detail the physiological function of the alpha-synuclein protein. Although discovered 30 years ago, the precise physiological function of α-Syn has not yet been fully unraveled. It is localized in the brain, mainly in neuronal presynaptic terminals, bound to the membranes of synaptic vesicles. In addition, α-Syn has been found in the nucleus, endoplasmic reticulum, Golgi apparatus and endolysosomal system of neurons, although in lower concentrations than those found in synaptic locations and its function is even less well understood. In this review, we would like to highlight the α-Syn function as a central protein in the regulation of synaptic events, and have focused in detail on its role in the regulation of serotonergic neurotransmission.
We agree with the Reviewer that α-Syn protein is characterized by remarkable structural dynamics, which depending on the environment, i.e., neighboring proteins, lipid membranes, redox state, local pH, interactions with ligands (ions, polyanions, and polycations). α-Syn adopts a monomeric, random coil conformation in an aqueous solution, while its interaction with lipid membranes drives the transition of the molecule part into an α-helical structure. The central unstructured region of α-Syn is involved in fibril formation by converting to well-defined, β-sheet rich secondary structures. These structural and biophysical properties probably hold the key to their normal and abnormal function. We have now added this information in the text (Pag 4). However, a detailed characterization of the mechanisms involved in these processes is beyond the scope of this review.
2- The authors should describe the difference between physiological α-Syn and pathological α-Syn(mutant α-Syn) in detail.
We appreciate the reviewer's great interest in the physiological and pathological mechanisms of α-Syn, and even mutated forms of α-Syn. However, this goes beyond the scope of the review, which is to give a current view of the role of α-Syn in the serotonergic system. In this context, we delve into the physiological/pathological aspects of α-Syn reported in humans and animal models of α-synucleinopathy.
Reviewer 2 Report
The reviewer thanks to authors for an excellent review of the literature and has no comments.
Author Response
We thank this reviewer for his/her positive comments about the study.
Reviewer 3 Report
The author summarize the linkage between synuclein and 5-HT system, which is interesting to PD. However, besides discussion in the present manuscript, the author should set up a section to discuss on the pathological role of 5-HT and its metabolites on synclein pathogenesis. The following papers should be cited and discussed (The neurotransmitter serotonin interrupts α-synuclein amyloid maturation. Biochim Biophys Acta. 2011 May; 1814(5): 553–561. The serotonin aldehyde, 5-HIAL, oligomerizes alpha-synuclein. Neurosci Lett. 2015 Mar 17; 590: 134–137.)
Author Response
We appreciate the reviewer's suggestions. We have now added these references and discussed them in the context of the complex relationships between α-Syn and the serotonergic system (Pag 9).
Reviewer 4 Report
mention of 5-HT enhancing anti-parkinson effects could be made.
limitations?
5-HT - dopamine interactions?
Author Response
We appreciate the reviewer's suggestions. The interaction between dopamine and 5-HT neurons has been evidenced in the mechanism of action of L-DOPA. The ectopic release of dopamine (DA) from 5-HT neurons and the clearance of extracellular DA by the norepinephrine transporter in areas enriched with noradrenergic terminals contribute to extracellular DA produced by L-DOPA and offer opportunities to improve anti-parkinson therapy. In fact, ample evidence suggests an interplay between DA and 5-HT in the appearance of L-DOPA-induced dyskinesia (LID), the most troublesome side effect of L-DOPA therapy, and some 5-HT1 receptor agonist may provide therapeutic benefits (1-4). However, a complete review of the brain 5-HT-DA interactions is outside the scope of the present article.
Some references
- Cohen SR, Terry ML, Coyle M, et al. The multimodal serotonin compound Vilazodone alone, but not combined with the glutamate antagonist Amantadine, reduces l-DOPA-induced dyskinesia in hemiparkinsonian rats. Pharmacol Biochem Behav. 2022;217:173393.
- Cohen SR, Terry ML, Coyle M, et al. The multimodal serotonin compound Vilazodone alone, but not combined with the glutamate antagonist Amantadine, reduces l-DOPA-induced dyskinesia in hemiparkinsonian rats. Pharmacol Biochem Behav. 2022;217:173393.
- Corsi S, Stancampiano R, Carta M. Serotonin/dopamine interaction in the induction and maintenance of L-DOPA-induced dyskinesia: An update. Prog Brain Res. 2021
- Sgambato V. Breathing new life into neurotoxic-based monkey models of Parkinson's disease to study the complex biological interplay between serotonin and dopamine. Prog Brain Res. 2021;261:265-285.
Reviewer 5 Report
Dear Editor,
I reviewed the manuscript by Miquel-Rio et al., entitled “ The Role of α-Synuclein in the Regulation of Serotonin System: Physiological and Pathological Features “
This review is very interesting and provides information about α-Syn and their role in regulating the 5-HT function in health and disease. In addition it is well written and and pictures and diagrams support of the context of the whole review. Accordingly, this review can be published in the present form.
Many thanks
Author Response

(The authors gave the same response as above.)
